# Spatiotemporal Mapping of Online Interest in Cannabis and Popular Psychedelics before and during the COVID-19 Pandemic in Poland

**DOI:** 10.3390/ijerph19116619

**Published:** 2022-05-29

**Authors:** Ahmed Al-Imam, Marek A. Motyka, Zuzanna Witulska, Manal Younus, Michał Michalak

**Affiliations:** 1Doctoral School, Poznan University of Medical Sciences, 60-512 Poznan, Poland; 2Department of Computer Science and Statistics, Poznan University of Medical Sciences, Rokietnicka 7 St. (1st Floor), 61-806 Poznan, Poland; michal@ump.edu.pl; 3Department of Anatomy, College of Medicine, University of Baghdad, Baghdad 10001, Iraq; 4Institute of Sociological Sciences, University of Rzeszow, 35-959 Rzeszów, Poland; mmotyka@ur.edu.pl; 5Faculty of Psychology and Law, SWPS University of Social Sciences and Humanities, Kutrzeby 10, 61-719 Poznan, Poland; zuzannawitulska@gmail.com; 6Iraqi Pharmacovigilance Centre, Ministry of Health, Baghdad 10001, Iraq; manalyounus@gmail.com

**Keywords:** entheogens, hallucinogens, Holt–Winters exponential smoothing, clustering analysis, social anomie theory, spatiotemporal mapping, time series analysis, web analytics

## Abstract

Background: Psychedelics represent a unique subset of psychoactive substances that can induce an aberrant state of consciousness principally via the neuronal 5-HT2A receptor. There is limited knowledge concerning the interest in these chemicals in Poland and how they changed during the pandemic. Nonetheless, these interests can be surveyed indirectly via the web. Objectives: We aim to conduct a spatial-temporal mapping of online information-seeking behavior concerning cannabis and the most popular psychedelics before and during the pandemic. Methods: We retrieved online information search data via Google Trends concerning twenty of the most popular psychedelics from 1 January 2017 to 1 January 2022 in Poland. We conducted Holt–Winters exponential smoothing for time series analysis to infer potential seasonality. We utilized hierarchical clustering analysis based on Ward’s method to find similarities of psychedelics’ interest within Poland’s voivodships before and during the pandemic. Results: Twelve (60%) psychedelics had significant seasonality; we proved that psilocybin and ayahuasca had annual seasonality (*p*-value = 0.0120 and *p* = 0.0003, respectively), and four substances—LSD, AL-LAD, DXM, and DOB—exhibited a half-yearly seasonality, while six psychedelics had a quarterly seasonal pattern, including cannabis, dronabinol, ergine, NBOMe, phencyclidine, and salvinorin A. Further, the pandemic influenced a significant positive change in the trends for three substances, including psilocybin, ergine, and DXM. Conclusions: Different seasonal patterns exist for psychedelics, and some might correlate with school breaks or holidays in Poland. The pandemic induced some changes in the temporal and spatial trends. The spatial-temporal trends could be valuable information to health authorities and policymakers responsible for monitoring and preventing addictions.

## 1. Introduction

New, or novel, psychoactive substances (NPS) are diverse—hundreds might exist today—and several schemes exist to categorize them [1]. The European Monitoring Centre for Drugs and Drug Addiction (EMCDDA) categorized NPS into cannabinoids, cannabimimetic substances, phenethylamines, cathinone derivatives, tryptamines, piperazines, pipradrol derivatives, and miscellaneous substances [2]. Many NPS possess a high-risk profile due to their relatively high incidence of intoxications and deaths. These substances are not limited to dimethoxyamphetamines (DMA/DOX), methoxetamine (MXE), mescaline, methylone, gamma-hydroxybutyrate (GHB), N-methoxybenzyl (NBOMe), a phenylethylamine derivative (2C-B), dimethyltryptamine (DMT), methamphetamine, lysergic acid diethylamide (LSD), and methylenedioxymethamphetamine (MDMA) [3]. Many of these chemicals are either psychedelic or can induce a dissociative state of consciousness, while some medicinal and research chemicals (RCs) with psychoactive or psychedelic properties are valuable as pharmaceuticals in medicine, such as ketamine, cannabis, and LSD [4,5]. Relevant authorities should monitor, regulate, and control these drugs with vigilance; hence, pharmacovigilance is the science and activities of detecting, assessing, understanding, and preventing adverse effects or other drug-related problems [3,6]. Pharmacovigilance expanded in the 1960s following the thalidomide tragedy, and for many drug safety interventions, such as drug withdrawals, labeling changes, and prescription restrictions, it showed an exponential progression and importance in health programs; this has led, over the last decade, to a fast evolution of regulation with more requirements from health authorities [7].

Psychedelic chemicals, also known as hallucinogens or entheogens, exist within the pharmaceutical “family” of psychoactive substances [8]. Psychedelics induce aberrant states of consciousness—principally via an agonist effect on a specific receptor of monoamine neurotransmitter—known as the 5-hydroxytryptamine type 2a (5-HT2A) receptor [4]. Psychedelics users also refer to these chemicals as spiritual aids, mysticomimetics, or psychotomimetics [3]. The public’s interest, including online interest—known as the online information-seeking/searching behavior—in psychedelic chemicals and actual substance (mis)use is still ambiguous [9]. Previous studies indicated that it could vary based on geographic location, time, and other demographics, including socio-economic parameters [3,5]. There are specific psychedelics of reemerging interest in psychiatry and neuroscience, including psilocybin, ayahuasca, DMT, LSD, and other synthesized medicinal chemicals that possess hallucinogenic properties, such as ketamine. There has been some recent evidence—based on randomized controlled trials (RCTs)—on the therapeutic effectiveness of those chemicals at the micro-dose range to manage debilitating conditions, including cognitive impairment, Alzheimer’s disease, and refractory psychiatric entities within the neurotic spectrum, including the post-traumatic stress disorder (PTSD), recalcitrant obsessive-compulsive disorder (OCD), panic disorder, and refractory depression [10,11,12,13].

The internet took over the function of a black market for illicit and controlled drugs; on websites, drug users have the opportunity to acquire all information about psychedelics, their use, risks associated with use, and the possibility of purchase although these online portals aimed at users of these drugs echo ignorance and carelessness, and they are devoid of references to objective scientific knowledge [14]. The 2020 report on the state of drug addiction in Poland indicated that the illicit drug market did not show a decrease in the availability of those chemicals in the first months of the pandemic; there were local restrictions, e.g., in Krakow, but drug prices did not change. Furthermore, the authors indicated that the role of the internet as a source of psychoactive drug supply may have increased [15,16].

There have been no studies to evaluate the spatial-temporal (spatiotemporal) mapping of online information-seeking behavior concerning psychedelics in Poland. The current study focuses on the most popular psychedelics. We explored that via time-series analysis because we expect repetitive seasonal patterns (periodicity). Our primary objective was to explore the spatial (geographic) and temporal (chronological) mapping of online information search behavior concerning psychedelics by surveying the trends databases across the world wide web. Our web queries’ (trends’) study scrutinized the interest of web users concerning the most popular hallucinogens (psychedelics) and akin web queries. The present study incorporates high-dimensional data analysis to yield the spatial (geographic) and temporal (time series) mapping concerning cannabis psychedelics before and during the pandemic in Poland.

## 2. Materials and Methods

### 2.1. Google Trends Data

We used Google Trends to retrieve raw data on the online information search (seeking) behavior concerning popular psychedelics in Poland. Microsoft Google launched the Google Trends website in 2006; it analyzes the popularity of top web search queries—via the Google search engine—of hundreds of millions of internet users [17], and here, we are referring to Google Trends data surveying the whole internet, i.e., the whole world with an estimate of the population around 7.753 billion (2020 census data). On the other side, the number of people in Poland who are interested in psychedelics varied from one substance to the other, and according to that, we ranked the popularity of these psychedelics from the most popular to the least popular based on substances that achieved the top (highest) web search queries.

Google Trends conveys data on four principal domains, including the interest by time (temporal or chronological mapping, i.e., time series), the interest by region (spatial or geographic mapping), related search topics, and related queries across the Internet [18,19]. We retrieved retrospective data for five years from the 1 January 2017 to the 1 January 2022 at weekly intervals creating a time series with 261 data points (weeks) for each substance. The total period encompassed the pre-pandemic with 166 data points and the pandemic era with 95 data points; the division of the time series relates to the time of discovery of patient zero in Poland on the 4 March 2020, which took place at the data entry point number 166 (week number 166).

Google Trends imposed a weekly-based temporal resolution, i.e., the default raw data from Google Trends are extractable (downloadable) at weekly intervals. Albeit we could transform it into monthly or quarterly data, that would be detrimental to our analytics—specifically the Holt–Winters exponential smoothing because we need a relatively high temporal resolution, i.e., smaller time units. On the other side, deploying daily time intervals would render the time series analytic exhaustive, lengthy, and redundant. Therefore, we abided by the default temporal resolution of Google Trends (weekly intervals). Further, week-by-week temporal resolution permitted the existence of sufficient data points (261); these are crucial for reliable and valid time series analysis and effectively comparing the pre-pandemic versus the pandemic eras.

Concerning the study sample, our research falls under info-demiology (information epidemiology) and info-veillance (information surveillance) studies, and we designed the current study upon a cornerstone time series analysis. The time series relates to the online information search behavior within the entire population of Poland, which is approximately 40 million (2020 census data). Therefore, the maximum sample size can never exceed that; however, the sample should relate to individuals who have access to the Internet and explore information concerning cannabis and psychedelics.

### 2.2. Pilot Exploration of the Trends

We explored the peer-reviewed literature, online drug fora, and websites dedicated to psychedelics and psychedelics enthusiasts to identify the most popular of these chemicals [8,20,21,22,23,24]. We created a preliminary list of twenty-five psychedelics—regardless of the ranking of popularity—including 2C-B, 6-allyl-6-nor-LSD (AL-LAD), alpha-Methyltryptamine, ayahuasca, datura, DMT, dimethoxybromoamphetamine (DOB), 2,5-Dimethoxy-4-methylamphetamine (DOM), dronabinol, dextromethorphan (DXM), ecstasy, ergine, GHB, ibogaine, ketamine, LSD, marijuana, mescaline, methamphetamine, nabilone, NBOMe, phencyclidine, psilocybin, salvinorin A, and 2,4,6-trimethoxyamphetamine (TMA-6). We utilized the list for an initial (pilot) exploration of Google Trends for five years retrospectively, including the pre-pandemic and the pandemic era.

### 2.3. Region-Specific “Dictionaries” of the Most Popular Psychedelics

We explored the spatial-temporal mapping for all 25 psychedelics by creating a “dictionary” specific to Poland that encompasses search terms for each psychedelic. The dictionary included the chemical name, scientific name, commercial or brand names (when applicable), whole and abbreviated names, street names, Polish-translated names, and related biological plant or mushroom species names from which the psychedelic can be extracted while excluding non-specific (non-yielding) terms from the dictionary. The dictionary included cannabis (marijuana) because: (1) Some psychiatrists believe it possesses psychedelic properties; (2) several psychedelics enthusiasts, users, and websites included it as a psychedelic substance; and (3) further, the United Nations dedicated an entire chapter of their World Drug Report (2019) to psychedelics, including cannabis—the 5th chapter (booklet) by the title “Cannabis and Hallucinogens” [25]. To achieve the highest and most specific search results via Google Trends, we combined the former categories of terms for each psychedelic using quotation marks, Boolean operators, and truncations, as shown in the Appendix A.

### 2.4. Statistical Analysis and Ethics

The research relied on raw data acquisition and subsequent multifaceted data analysis. We utilized frequentist statistics and times series analysis using an additive model of Holt–Winters Exponential Smoothing to detect potential seasonal patterns of periodicity for the online information-seeking behavior [26]. We evaluated the best model for time series analysis by utilizing the mean absolute error (MAE). Additionally, we compared the slopes of linear trends (pre-pandemic versus pandemic) to check if there was a significant change in online interest in psychedelics for those two periods. Further, based on Ward’s method (criterion), we implemented hierarchical clustering analysis concerning the geographic mapping of online behavior within Poland. We ran the statistical analysis using JMP Pro v.16 (time series analysis) and Statistica v.13.3 (spatial mapping and clustering analysis). All tests were considered significant at a *p*-value < 0.05. The current study did not require ethical approval because it did not involve patients; further, we worked on publicly available (open-access) data on the Internet.

## 3. Results

According to the Holt–Winters exponential smoothing, we detected significant seasonal patterns concerning 60% of the psychedelics, while there was no obvious seasonal pattern for DMT, MDMA, ketamine, mescaline, methamphetamine, ibogaine, 2C-B-FLY, and nabilone (Table 1). The remaining psychedelics have different seasonal patterns; for instance, mushroom-related substances (psilocybin and ayahuasca) had a significant annual seasonality every 49 weeks (estimate = 0.322 +/− standard error of estimate = 0.127, *p*-value = 0.0120) and 52 weeks (0.414 +/− 0.113, *p* = 0.0003), respectively. Psilocybin and ayahuasca had an interesting periodic pattern; they always reached the time series’ highest levels during September and October of each year; this pattern coincides with the seasonal growth and harvest time of mushrooms and magic mushrooms in Poland [27,28,29]. Nonetheless, the former assumption might be true concerning psilocybin only given that components (ingredients) for ayahuasca are not grown in Poland and do not correspond to the harvest season; therefore, the reason for the annual seasonality of ayahuasca is unknown and requires further study.

Four substances—LSD, DOB, DXM, and AL-LAD—had a bi-quarterly seasonality (i.e., every half a year) at 27 (0.194 +/− 0.086, *p* = 0.0255), 24 (0.194 +/− 0.081, *p* = 0.0169), 28 (0.221 +/− 0.079, *p* = 0.0056), and 24 (0.232 +/− 0.066, *p* = 0.0004) weekly intervals, respectively. The remaining six psychedelics had a significant quarterly seasonal pattern; the most significant pattern was for NBOMe at a 99.99% confidence interval (13 weeks, 0.386 +/− 0.066, *p* < 0.0001). The remaining five substances included cannabis (19 weeks), phencyclidine (14 weeks), ergine (14 weeks), salvinorin A (10 weeks), and dronabinol (17 weeks); the quarterly periodic pattern may correlate with leisure time during school breaks and relevant holidays.

Concerning other systematic components, i.e., level and trend, of the time series, none of the substances had a significant trend component, while only five psychedelics exhibited a significant level, including NBOMe, cannabis, LSD, phencyclidine, and DOB. We estimated the MAE for each Winters exponential smoothing model; it was lowest for NBOMe (4.629) and highest for ayahuasca (23.703); the remaining psychedelics ranked in between, including cannabis, AL-LAD, phencyclidine, LSD, salvinorin A, dronabinol, ergine, psilocybin, DXM, and DOB. To summarize, almost two-thirds of the surveyed psychedelics exhibited seasonal periodicity; nonetheless, the seasonal pattern was heterogeneous, and most synthetic psychedelics had either quarterly or half-yearly seasonality or annual seasonal pattern.

We additionally analyzed the slopes (trends) of the time series for each psychedelic for the pre-pandemic versus the pandemic period in Poland, as seen in the Appendix A. There was no tangible difference between the trends before and during the pandemic. However, we observed a minor increment for 11 psychedelics, including cannabis, ketamine, mescaline, psilocybin, DOB, DXM, ergine, dronabinol, 2C-B-FLY, nabilone, and AL-LAD. At the same time, the remaining nine psychedelics had a comparable change but in the opposite direction of change in trends. Nonetheless, only seven substances had a significant change in the slope over the two periods; three psychedelics had incremental change, including psilocybin (−0.1676, *p* = 0.030), ergine (−0.2261, *p* = 0.005), and DXM (−0.2302, *p* = 0.005). The remaining four had a decremental change, including NBOMe (coefficient = 0.1327, *p* < 0.001), LSD (0.2421, *p* < 0.001), MDMA (0.2160, *p* < 0.001), and phencyclidine (0.1819, *p* = 0.006) (Table 2).

Concerning the spatial mapping for the total period in Poland, the hierarchical clustering conveyed two clusters; the smallest had six voivodships, including the greater Poland, lesser Poland, Lower Silesian, Masovian, Pomeranian, and Silesian voivodship. The second cluster had the remaining (ten) voivodships. Clustering analysis during the pre-pandemic period also generated two clusters; the largest had lesser Poland, Lublin, Lubusz, Opole, Podkarpackie, Podlaskie, Swietokrzyskie, Warmian-Masurian, and west Pomeranian, while the other cluster had remaining (seven) voivodships (Figure 1). On the other side of the time series (pandemic period), hierarchical clustering yielded not two but three clusters; the smallest had three voivodships, including Opole, Warmian-Masurian, and Lubusz. The largest had greater Poland, lesser Poland, Lublin, Podkarpackie, Podlaskie, and west Pomeranian, while the third cluster had the remaining (six) voivodships (Figure 2). To summarize, clustering generated two clusters for each of the summative (total) and pre-pandemic periods and three distinct clusters during the pandemic.

Concerning our study, the highest interest was towards NBOMe; search indications for this specific substance were highest in all regions of Poland before and during the pandemic. Similarly, before the pandemic and during its first months, cannabis derivatives were frequently searched in almost all provinces; however, low search interest included Opolskie (pre-pandemic) and Lubuskie, Opolskie, and Warmińsko-Mazurskie provinces (post-pandemic). Variations existed for MDMA; however, our results show an increase in search during the pandemic almost all over Poland. As noted in the present research concerning LSD, increased interest was evident during the pandemic. No differences existed in the search for information on DMTs before and during the pandemic; nonetheless, in four provinces (Świętokrzyskie, Podlaskie, Opolskie, and Lubuskie), the interest in this agent was similar to that in the pre-pandemic, while in the remaining regions, interest in this drug was relatively high. In most provinces, some psychedelics are either extracted from mushrooms or plants (ibogaine, psilocybin, and ergine) and those produced by synthesis or extracted from non-prescription or over-the-counter drugs (ketamine, DOB, DXM, among others); there was a low search interest on online sites in both periods. In our study, we noted that the lowest interest among Internet users was for ibogaine.

## 4. Discussion

The purpose of the present study was to explore the temporal and spatial mapping of online information-seeking behavior concerning cannabis and psychedelics while conducting inferential time series analysis and examine if the pandemic influenced distinctive temporal-spatial patterns before versus during the pandemic. We opine that online search behavior could correlate with real-life behavior, or it can be anticipatory (predictive) for it. Therefore, the results we inferred can be of value for policymakers, legislators, and health authorities to prognosticate, mobilize human and technical resources, and antagonize a projected change in actual world behavior that favors the purchase and use of those illicit chemicals. The preemptive measures might be critical from an economic and healthcare perspective to promote better societal health while not burdening the healthcare infrastructure and resources during overwhelming addiction crises.

The Internet is currently the primary source of all knowledge regarding psychedelics: knowledge both scientific and, above all, acquired through one’s own experiences with those substances as described in numerous websites run and directed to users of these drugs [30]. The information available on the web includes detailed descriptions of psychedelics, instructions on production, sources of acquisition, the physiologic and psychedelic effects that these drugs produce, legal aspects, education on how to prevent the occurrence of dangerous consequences in users of these drugs, and all kinds of relevant information [31]. Therefore, it is not surprising that people interested in this topic consider the Internet a valuable source of knowledge in this area.

Concerning our study, the highest interest was towards the phenylethylamine derivative NBOMe (which acts similarly to LSD); it was already of great interest among users of psychedelics in 2014 [32]. Drugs in this group are usually sold in blotter or powder form under various names: “Smiles”, “N-Bombs”, “Solaris”, “Cimbi” or “25I”, “25B”, and “25C” [33]. These agents have drawn the attention of researchers due to their established position on the drug market both in Poland [34] and their widespread use in other regions of the world [33], causing numerous fatalities [35]. Before and during the pandemic, search indications for this specific substance were highest in all regions. The increase in popularity of NBOMe since before the pandemic might be due to the unclear legal situation concerning its availability on the European drug market and its increasing popularity for almost a decade, probably caused by the dissemination of information about its appearance, effects, and availability through online portals targeting users of these drugs [36,37].

Similarly, in the period preceding the pandemic and during its first months, information on cannabis derivatives was equally frequently searched in almost all provinces. According to research data, drugs from the cannabis group are the most popular among Polish users, so the high search indications correspond with data from user studies [15]. The provinces where low interest in drugs from this group included Opolskie (pre-pandemic) in addition to Lubuskie, Opolskie, and Warmińsko-Mazurskie provinces (post-pandemic); these results are in harmony with previous studies, wherein earlier years, low or marginal percentages of people admitted using the substance [38].

For another famous psychedelic, MDMA, some variation was evident in search frequency before and during the pandemic. We emphasize that in 2018–2019, indications of the use of MDMA suggested its low popularity; researchers estimated it at 0.2% in the general population (15–64 age group); further, researchers observed a decline in interest in MDMA in recent years [15]. In studies conducted during the pandemic, almost two-thirds (59%) of study participants admitted to using MDMA last year [39]. However, we cannot compare these data due to the heterogeneous sampling. Our results show an increase in the search for MDMA during the pandemic in all provinces, which may or may not correspond with data from the cited studies.

In the years preceding the pandemic, estimates of LSD use in the Polish population aged 15–64 indicated 0.5% of users [15]. During the pandemic (in November 2020), using the Computer-Assisted Web Interview (CAWI) technique, investigators deployed a survey targeting psychoactive drugs’ users; they distributed the survey via Facebook and Instagram, in its dissemination in a sample suitable for this research, and activists of the Social Initiative for Drug Policy and the Polish Network for Drug Policy, among others, participated. They collected 2373 correctly completed questionnaires and obtained consent to use the results; these surveys showed that nearly four in ten people had used LSD in the past year, suggesting an incremental interest in LSD use during a pandemic [39].

No differences existed in the search for information on DMTs in periods before and during the pandemic. In four provinces (Świętokrzyskie, Podlaskie, Opolskie, and Lubuskie), the interest in this agent was similar to the period before the pandemic. In the remaining regions, interest in this drug was relatively high. The interest in this psychedelic occurs not necessarily for its psychedelic properties, which persist for 30–60 min after ingestion, but rather for its use when combined with more complex psychedelic concoctions for a multi-hour psychedelic trip [40].

In most provinces, for some psychedelics, both extracted from mushrooms or plants (ibogaine, psilocybin, and ergine) and produced by synthesis or extracted from over-the-counter (non-prescription) drugs (ketamine, DOB, DXM, among others), there was low search interest on online sites in both periods. The collected results may confirm the low popularity of these drugs due to the not very pleasant side effects of use (e.g., vomiting, nausea, and torpor) [41].

Our study shows that the lowest interest among Internet users was for ibogaine, which, although included in the group of psychedelics, physicians more often recommended to manage opioid addiction, including opioid withdrawal syndrome [42]; hence, it is likely that there were low indications of searching for information about this psychedelic in most provinces in both periods.

Considering that the search scale for psychedelics due to the selected province can also be referred to previous measurements, which partly correspond to the presented results. During the Social Diagnosis 2015 measurement, a relatively low percentage of respondents (about 0.7%) confirmed drug use existed in several indicated regions (Świętokrzyskie, Lubelskie, and Podkarpackie) compared to others [38]. These provinces—before the pandemic—also had some of the lowest drug overdose mortality rates (0%, 0.09%, and 0.38%) [15].

The pandemic represents a state of social anomie, as its dynamic development led to restrictions and new regulations worldwide; these restrictions made many people fear the possibility of illness and death, economic crisis, and the uncertainty of the future [43], but they also adopted attitudes of rebellion or rejecting the possibility of danger [44]. Researchers also pointed to the possibility of interaction anomie arising from the limitations of interpersonal relationships while at the same time looking for ways not to be isolated after all [45,46]. Psychological stressors associated with the pandemic, influencing the confusion and focus on survival and avoidance of the disease, were primarily the strong impact of mass and social media reporting constantly on cases and deaths, new mutations of the virus, lack of vaccine (in the first year of the pandemic), the threat of economic recession, unemployment, and reduced quality of life. These caused severe anxiety and depressive states and even led to post-traumatic stress disorder (PTSD) [47,48,49]. Some may try to take advantage of the new circumstances to profit from trading rare goods. The pandemic, particularly the experiences and behaviors that accompany it, has led to the disintegration of existing rules within society while introducing new rules necessary to achieve new goals: survival, avoidance of infection, and maintenance of social order [50,51]. Concerning responses to anomie, we hypothesized that specific individuals might be more interested in searching web-based information concerning psychedelics, including teenagers and young college students, because actual drug use is a behavior frequently seen among adolescents and young adults per the World Drug Report [25].

Some survey data confirm that drug use increased during the pandemic [52,53]. Some data, however, indicate a decrease in use, explained by, among other things, less influence from the peer environment [54]. Research suggests that interest in psychedelics during the COVID-19 pandemic was related to the need to reduce depressive states, anxiety, and insomnia, with the study authors reporting no significant change in trends in psychedelic use [55]. On the other side, the pandemic period has had a significant impact on the increase in Internet activity globally, and spending time on the Internet has ceased to be one of the daily choices and has somewhat become a necessary compulsion [56]. For those interested in psychedelics, the Internet has proven valuable in acquiring the necessary information and a platform to trade psychedelics [57,58]. Further, researchers have highlighted the potential for using Internet-based drug search data to predict public health events; these data are cost-effective sources of insights because they are free, publicly available for research, and provide the opportunity to predict consumer behavior for these drugs [59].

Zaami et al. (2020) speculate that some people seeking access to psychedelics—due to impeded access in the real world—may have switched to seeking these drugs via online sites [60]. Groshkova et al. (2020) partially confirmed the assumptions mentioned before. During the first three months of the pandemic, researchers identified a 27% increase in sales of cannabis derivatives on darknet markets. The authors speculated that part of the drug market may have moved online, including encrypted darknet sites [58]. Other researchers at the European Monitoring Centre for Drugs and Drug Addiction have also confirmed higher Internet activity levels related to psychedelics’ online sale, especially cannabis derivatives and stimulants [57]. Researchers of this phenomenon observed mixed trends in psychoactive substance use. A study concerning German psychoactive drug users found that the pandemic had little effect on changing trends of substance access and use; only 2% of the sample obtained drugs online [61].

The authors of the European Drug Report consider that the drug market appeared to be very resilient to any pandemic-related difficulties, indicating at the same time that drug traffickers were quick to adapt to any pandemic-enforced constraints and lockdowns, e.g., by making use of Internet-based distribution options. One of the concerns raised by the report’s authors is that distribution channels for psychoactive drugs may remain in the virtual space for a long time. Available information allows us to note that drug consumption declined in the initial period of the pandemic but quickly returned to previous indications once pandemic-related restrictions were no longer obligatory. At the pandemic’s beginning, the decline was particularly in “recreational” drugs, including MDMA and amphetamines, while interest increased in psychedelics, including LSD, and substances with dissociative properties, such as ketamine, seen as more appropriate for use at home [62]. The ability to search the Internet for information on these drugs relates to our research.

The increase in popularity of some psychedelics in many regions of Poland may be related to the information appearing in opinion-forming magazines conveying experiences after this drug as mystical, instructive experiences [63,64]. In all likelihood, attitudes towards psychedelics can manifest in correspondence to mass culture products filled with numerous liberal descriptions, which can arouse curiosity and encourage people to try a particular drug [65]. It is also possible that the increase in searches for some psychedelics observed online during the pandemic was related to the need to seek help on the websites of counseling and support services for users of these drugs, where remote addiction treatment, harm reduction, and counseling services were made available during the pandemic [66]. These data, however, also require detailed verification.

In 2019 and 2022, Motyka and Al-Imam studied the etiology of psychoactive substances and psychedelic use; they have also investigated the relationship between musical preference and views on the use of drugs among youth and debated the representation of these substances in mass culture may influence the liberalization of attitudes towards their use [65,67,68]. Further, the representation of psychoactive drugs—including psychedelics—in mass culture may influence the liberalization of interest or attitudes towards drugs and their use; mass culture relates to films, music, literature, everyday products, e.g., food, clothes, and cosmetics, in addition to role models, including actors, musicians, and celebrities [65]. However, psychedelics enthusiasts also declared a specific interest in those chemicals other than adolescents and young adults, for instance, using psychedelics for a planned pre-mortem psychedelic trip in the elderly and those with debilitating conditions, including malignant tumors; most psychedelic users recommended using DMT before death, while others suggested tripping on LSD, psilocybin and psilocybin mushrooms, and NBOMe compounds [69]. Researchers also confirmed that psychedelic users have tendencies to abuse high authority [70].

During the pandemic, the augmented role of the Internet to search for psychedelics and their exponential abuse represents distinct social attributes; these attributes are also evident in EU members other than Poland and other regions of the world [60,61,71,72,73]. The temporary closure of borders due to the expanding COVID-19 pandemic was one of the reasons why the distribution of psychedelics shifted to online networks. The scarcity of classic street drugs was made difficult by limited transport between countries, the closure of entertainment centers (where users can purchase these drugs), and the prolonged need to stay indoors during the lockdown are other important determinants of the increased demand for different accessibility to psychedelics than before the pandemic. These phenomena have been observed, among others, in large cities in Northern Europe [60]. In a study conducted during the pandemic among Georgian psychoactive drug users, researchers found that one in four participants (25%) bought their drugs via the Internet [71]. During the first months of the pandemic, increased intensity of drug transactions on cryptomarkets was observed [72]. In addition, on the English-language Internet platform Reddit, which provides users with complete anonymity, already in the first six months of the pandemic, there was a significant increase in posts concerning any information related to the use of psychedelics: trade, production, use, and any other information related to this topic [73]. A survey of German psychedelic users noted that a shift to substitutes for known drugs bought through online sites might result from the gaps in existing access caused by pandemic restrictions. While this study did not find an increase in online drug transactions [61], as noted above, searching the Internet for information related to psychedelics does not necessarily imply a desire to purchase them. Surveys of psychedelic users have confirmed that some of them have used Internet-based services for, among other things, harm reduction resulting from the use of psychoactive substances and the difficult access to traditional forms of treatment during the pandemic [74].

Although the situation in Poland does not differ significantly from the results observed in other countries in the world, the results of the presented studies may confirm the observed trends in the growing importance of the Internet in searching for any information on psychedelics.

The results established in the presented research on the temporal and spatial variation of Internet searches for information on psychedelics—when made available to decision makers—can be used to determine appropriate strategies and countermeasures. The Ministry of Health can consider the results and design preventive interventions that integrate into the National Health Programs to prevent and treat addiction. In addition, the provision of data on the periodicity of psychedelic use may allow more precise planning of drug prevention and harm-reduction policies. If the work is valuable, it can be shared and adapted to design drug strategies in other regions of Europe and the world. Our methodology may apply in other similar studies, which may not only prove to be studies that test our methods but, in our opinion, may contribute to strengthening drug prevention interventions.

### 4.1. Study Limitations

Our research has limitations inherent to the observational study design, including having a lower level of evidence than experimental or quasi-experimental studies, being prone to biases and confounding variables, and having a reduced capacity to infer causality. Additionally, web users who are psychedelics enthusiasts or (mis)users may prefer to be anonymous while surfing the web. Internet users can rely on Internet protocol (IP) masking, virtual private networks (VPN), VPN applications, or utilize web browsers and search engines dedicated to anonymous Internet use, including Tor browser and DuckDuckGo search engine [71,72,73]. Unfortunately, we could not account for multiple searches from the same IP address; this aspect also relates to IP masking issues and the use of VPN technologies. Besides, multiple searches from the same IP address may relate to the same individual or a group of individuals accessing the same IP address. Nevertheless, most Internet service providers (ISPs) assign a dynamic IP address rather than a static one. In addition, multiple searches from the same static IP address mean persistent or recurring interest in these chemicals, which represent an essential data signal that we should record and analyze rather than ignore.

Google Trends also possesses inherent limitations and restrictions; for instance, the search results available through Google Trends are anonymous and only reflect those with Internet access, potentially excluding specific groups of interest, for example, the elderly and underprivileged groups that lack access to the Internet in low- and middle-income countries [74]. Besides, Google Trends only conveys relative numbers at a percentile scale, and there is no official way to access the absolute numbers. Accordingly, Google Trends poses some restrictions on data scientists concerning data analytics. Google Trends also derive data from web users utilizing the Google search engine only, and therefore, web queries based on other search engines, including Baidu, DuckDuckGo, Ecosia, Dogpile, and WolframAlpha, among others, are uncharted (non-mappable) via Google Trends [75,76,77,78].

Concerning the present study, we could not conduct a time series analysis for the pre-pandemic and pandemic periods separately; this relates to the relatively short pandemic period that is not suitable for analyzing seasonal patterns. For instance, psilocybin and ayahuasca have annual seasonality; therefore, we need multiple years within the pandemic to confirm if the periodicity may persist or change to an alternative pattern. Nonetheless, the former assumption may not be accurate because as the pandemic progressed, people became less alarmed and reverted to routine activities, rendering Google Trends data of less value concerning interest in psychedelics. The pandemic’s panic lessened significantly due to the habituation (adaptation) to the new social distance measures, the advent of effective vaccines, and specific therapeutics for the COVID-19.

### 4.2. Future Research

Future studies can aim to collect data from the Polish National Health Fund (NFZ: Narodowy Fundusz Zdrowia), toxicology units, hospitals, emergency units, and potentially from the police units in Poland. The former is congruous with the importance of gathering data from the voivodships on drug overdoses, incarcerations, and hospital admissions. Further, future research should focus on health-related metrics and socio-economic parameters or indices. We analyzed and compared those metrics with Google’s online interest data in the current study. For instance, we analyzed the human development index (HDI), the gross regional product (GRP) for each voivodship, the population count for each voivodship, and finally the online interest in the deep web. We conducted correlations with Google’s online interest data; we found some positive and significant results.

Nonetheless, according to the calculations of the correlation coefficient, most of these correlations possessed a small (weak) effect size (Appendix A); moreover, “correlation does not imply causation”. Further, we implemented the correlations for the total period, the pre-pandemic era, and during the pandemic (Appendix A). We found no drastic changes concerning the correlation between different eras. However, when it comes to the online interest in the deep web (Appendix A), we conducted binary (logical) indexing to contrast the pre-pandemic versus the pandemic period. Concerning the correlations with online interest in the deep web, there were only a few differences, the majority of which were related to correlations with the GRP and interest in the deep web; most of the changes in correlations were concerning MDMA although others existed for NBOMe, DOB, DXM, ergine, and methamphetamine (Appendix A). Again, those results are positive but entail a weak (small) effect size. Besides, those socio-economic parameters are compared at the voivodship level—not at the individual level—and therefore, our analytics have a low level of evidence because they refer to aggregate data or meta-data, which may provide confusing conclusions concerning causality. Accordingly, we opined that the former results are relatively weak when integrated with other analytics, including the time series and the heatmap geographic analyses.

## 5. Conclusions

The study results show a distinctive pattern of interest in psychedelics in Poland. Temporal and spatial differentiation existed, and previous studies support our results. The added value is determining the periodicity of interest in particular psychedelics, which may be necessary for policy planning for drug addiction prevention.

## Figures and Tables

**Figure 1 ijerph-19-06619-f001:**
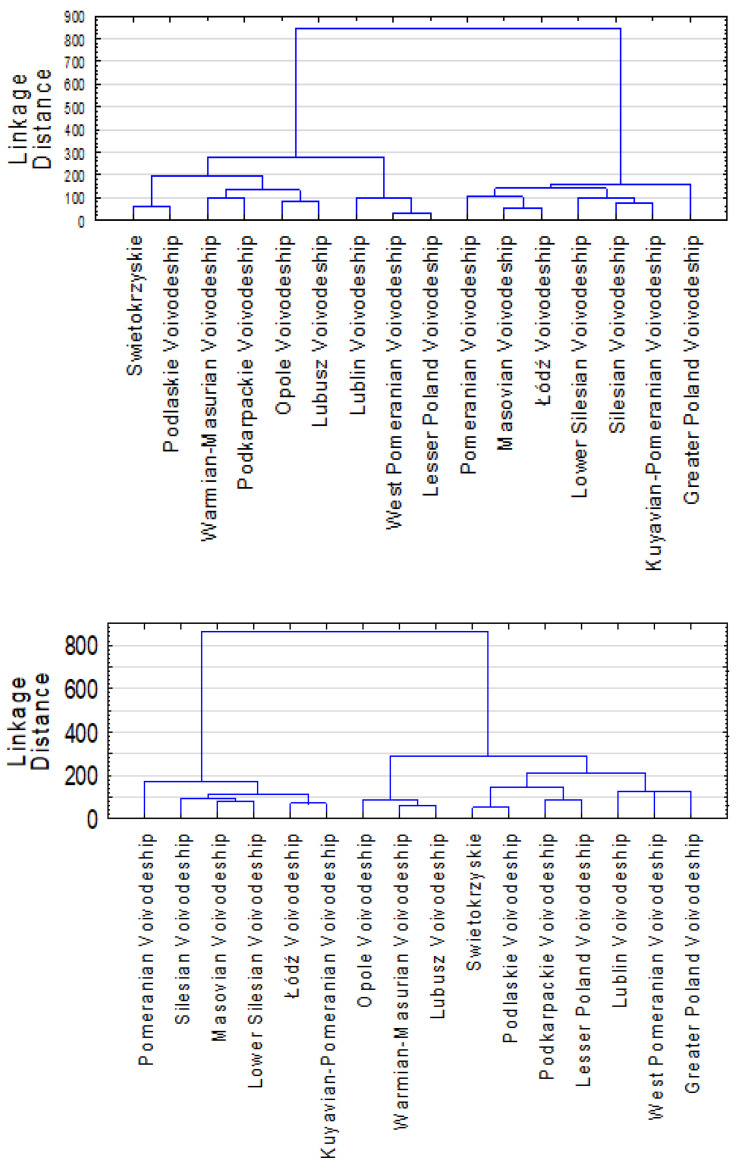
Poland’s hierarchical clustering for the pre-pandemic (**top**) and pandemic period (**bottom**).

**Figure 2 ijerph-19-06619-f002:**
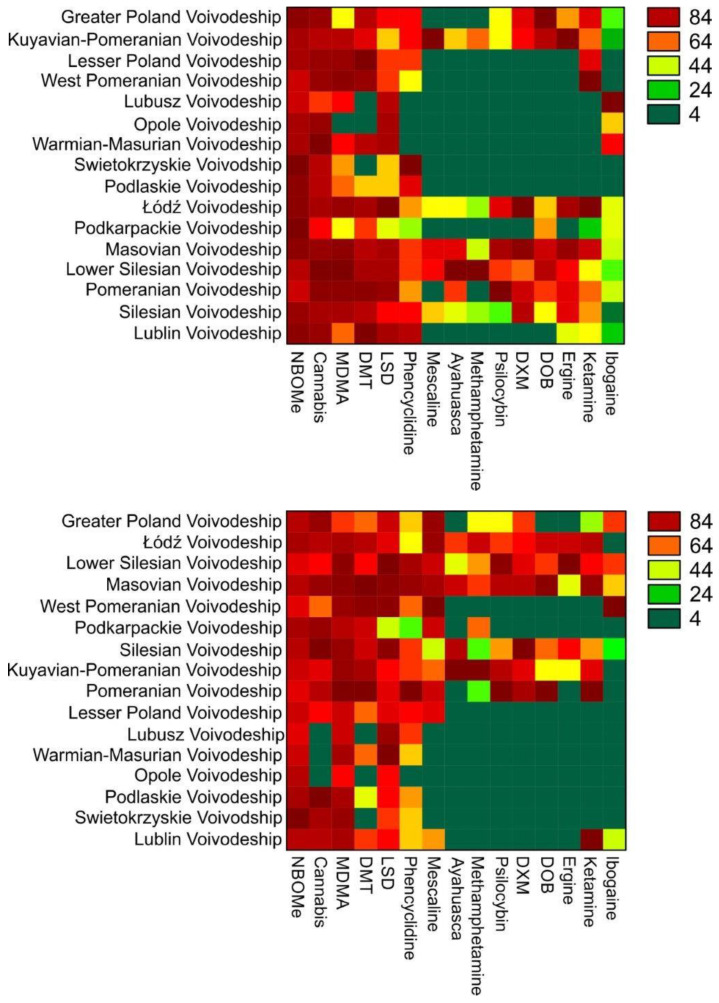
Poland’s spatial heat map for the pre-pandemic (**top**) and pandemic period (**bottom**).

**Table 1 ijerph-19-06619-t001:** Time series analysis for Poland: Holt–Winters exponential smoothing.

Psychedelic	OPP	Df	MAE	Level			Trend			Seasonality		
				Estimate	SE	*p*-Value	Estimate	SE	*p*-Value	Estimate	SE	*p*-Value
NBOMe	13	244	4.629	0.321	0.056	**<0.0001**	<0.001	<0.001	0.9232	0.386	0.066	**<0.0001**
Cannabis	19	238	8.187	0.137	0.053	**0.0097**	0.000	0.017	1.0000	0.179	0.080	**0.0267**
LSD	27	230	14.989	0.054	0.027	**0.0446**	0.005	0.026	0.8582	0.194	0.086	**0.0255**
Phencyclidine	14	243	14.270	0.074	0.030	**0.0157**	0.000	0.000	0.8122	0.137	0.063	**0.0321**
Psilocybin	49	208	17.360	0.003	0.007	0.6822	0.251	0.625	0.6881	0.322	0.127	**0.0120**
Ayahuasca	52	205	23.703	0.054	0.030	0.0740	0.002	0.016	0.8957	0.414	0.113	**0.0003**
DOB	24	233	22.137	0.053	0.022	**0.0180**	0.001	0.001	0.6539	0.194	0.081	**0.0169**
DXM	28	229	18.544	0.032	0.036	0.3800	0.018	0.021	0.3830	0.221	0.079	**0.0056**
Ergine	14	243	17.340	0.032	0.023	0.1661	0.012	0.031	0.7096	0.139	0.060	**0.0208**
Salvinorin A	10	247	16.506	0.001	0.001	0.5192	0.992	1.447	0.4937	0.108	0.048	**0.0247**
Dronabinol	17	240	16.930	0.019	0.032	0.5499	<0.001	0.031	1.0000	0.172	0.058	**0.0032**
AL-LAD	24	233	9.879	0.016	0.017	0.3760	0.007	0.015	0.6222	0.232	0.066	**0.0004**
DMT	n/a											
MDMA												
Ketamine												
Mescaline												
Methamphet.												
Ibogaine												
2C-B-FLY												
Nabilone												

Significant *p*-values are in bold font. Abbreviations: Df, degrees of freedom; MAE, mean absolute error; n/a, not applicable (no significant seasonality); OPP, observations-per-period (length of seasonality period); SE, standard error.

**Table 2 ijerph-19-06619-t002:** The slope difference in linear trends: Pre-pandemic versus pandemic period.

Psychedelic Substance	Coefficient	*p*-Value *
NBOMe	0.1327	**<0.001**
Cannabis	−0.0053	0.895
DMT	0.0292	0.497
LSD	0.2421	**<0.001**
MDMA	0.2160	**<0.001**
Phencyclidine	0.1819	**0.006**
Ketamine	−0.0623	0.420
Mescaline	−0.0233	0.744
Psilocybin	−0.1676	**0.030**
Ayahuasca	0.1273	0.171
DOB	−0.1165	0.245
DXM	−0.2302	**0.005**
Ergine	−0.2261	**0.005**
Methamphet.	0.0338	0.643
Salvinorin A	0.0588	0.455
Dronabinol	−0.0855	0.281
Ibogaine	0.0360	0.507
2C-B-FLY	0.0128	0.808
Nabilone	−0.0596	0.285
AL-LAD	−0.0944	0.183

***** Significant *p*-values are in bold fonts. Significance was calculated based on a *t*-test for the slope difference of two independent samples.

## Data Availability

All data are available upon request from the corresponding author for three years following the publication, pending justifiable requests.

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
