# Peer review of "Spatiotemporal Mapping of Online Interest in Cannabis and Popular Psychedelics before and during the COVID-19 Pandemic in Poland"

_ijerph, 2022, doi:10.3390/ijerph19116619_

Round 1

Reviewer 1 Report

This manuscript was interesting; however, it must be improved in some minor, as well as more major ways from this reviewer's perspective. 

1) stronger and more citations for claims made (not limited to Ines 263, 320, 324). 

2) What is the sample size? Hundreds of millions? How many people in Poland looked up one of the 25 psychedelics? This reviewer didn't see this clearly stated. Also, how did you account for multiple searches from same IP address?

3) You hypothesized "that 95 specific individuals might be more interested in searching web-based information concerning psychedelics, including teenagers and young college students, those interested in selling (commercing) or electronically selling (e-commercing) psychedelics, and others who might surf the web during their leisure time (weekends and holidays)." However, age was not measured, nor does the design allow age to be measured. I recommend clarifying this or leaving it out.

4) More discussion on what the implications of the results might be. How might this impact clinical, legal, or policy areas?  What is the purpose of this study? Just curiosity? Perhaps comparing the rates of internet searches for certain drugs to drug stats, overdoses, AD/ED visits for drug use, etc. This would strengthen this study. It is interesting, but the discussion should focus on what we can do with the results. What direction should a further study take?

Author Response

Dear Sir/Madam,

Please see the attachment; thank you.

Reviewer 2 Report

Were the psychedelics chosen also considered when looking at data from the voivodships on drug overdoses, incarcerations, and hospitalizations? Do the authors have access to data on overdoses, incarcerations, hospitalizations, mortality that could be used to look at the relationships between these outcomes and the data from google? Online posts and searches may often precede increases in health events such as overdoses and mortality.

Are there any health-related metrics available at the population level that may also be used to compare differences in interest of psychedelics and whether they may be associated with perceived benefits?

Were any analyses conducted that examined socio-economic factors that may have been related to the interest in psychedelics by voivodships?

Is it possible to analyze the data by counties (powiats) that have lower socioeconomic levels compared to others that have higher socioeconomic levels and whether there were differences in the searches?

Were country specific COVID-19 related events taken into account when looking at seasonality?

Justification for the use of weeks for time is lacking.

A greater discussion on how public health professionals/agencies/health departments/government may use this data would strengthen the manuscript.

Author Response

(The authors gave the same response as above.)

Reviewer 3 Report

Authors investigated online information-seeking behavior concerning 
cannabis and the most popular psychedelics before and during the pandemic by using temporal and spatial mapping. The research in question is interesting and the article is well written. 

Major suggestions:

  1. Please check your references, n.37 does not match to what is written in the discussion
  2. The hypothesis in the introduction “specific individuals might be more interested…” (lines 95-99) was not actually researched as there are no data on who conducted the research. I would suggest rewriting this sentence.
  3. The discussion section is rather long and speaks of variables in extenso which are not actually part of results of this research. Although interesting, an entire passus in discussion (lines 378-390) is not relevant for the results and it only retells other sources. The suggestion is to either remove it, or just mention it in a sentence or two.
  4. Study limitations – lines 435-438, although informative, this is not a limitation to your study and should be removed or written elsewhere.

Minor suggestions:

  1. Results - lines 215-225. It is easy to get lost between spatial clusters and voivodeships. I suggest you write in (Figure 1) in the end of sentences in the line 221, and (Figure 2) in the end of sentences in the line 225.
  2. Discussion - there are several sentences that are too long/unclear and would benefit if rewritten or possibly divided in two sentences. E.g. lines 273-276; 305-307; 310
  3. Lines 288-289, "Data from recent studies that implemented a purposive sampling indicated that almost four in ten had used this psychedelic in the last year [37]. " Please describe with few words those samples - young people? addicts? General population? (I wanted to look out by myself, but the reference listed under no. 37 does not mention anything about the use of psychedelics.)
  4. Line 399. The sentence is not clear – what/which social attributes?

Author Response

(The authors gave the same response as above.)
